# Pregnancy-Associated Takotsubo Syndrome: A Narrative Review of the Literature

**DOI:** 10.3390/jcm14072356

**Published:** 2025-03-29

**Authors:** Panagiotis Iliakis, Anna Pitsillidi, Nikolaos Pyrpyris, Christos Fragkoulis, Ioannis Leontsinis, Georgios Koutsopoulos, Emmanouil Mantzouranis, Stergios Soulaidopoulos, Alexandros Kasiakogias, Kyriakos Dimitriadis, Günter Karl Noé, Konstantinos Tsioufis

**Affiliations:** 1First Department of Cardiology, School of Medicine, National and Kapodistrian University of Athens, Hippokration General Hospital, 11527 Athens, Greece; npyrpyris@gmail.com (N.P.); christosfragoulis@yahoo.com (C.F.); giannisleontsinis@gmail.com (I.L.); mantzoup@gmail.com (E.M.); soulaidopoulos@hotmail.com (S.S.); akasiakogias@gmail.com (A.K.); dimitriadiskyr@yahoo.gr (K.D.); ktsioufis@gmail.com (K.T.); 2Department of OB/GYN, Rheinland Klinikum Dormagen, Dr.-Geldmacher-Straße 20, 41540 Dormagen, Germany; anna.pitsillidi@gmail.com (A.P.); guenter-karl.noe@rheinlandklinikum.de (G.K.N.)

**Keywords:** takotsubo syndrome, pregnancy, peripartum, coronary syndrome, peripartum cardiomyopathy

## Abstract

Takotsubo syndrome (TTS) is a clinical syndrome defined most typically by transient systolic dysfunction and dilatation of the apex of the left ventricle or other regional areas in the documented absence of obstructive coronary artery disease. Although more commonly presented in postmenopausal women, there are reports in the literature of TTS during the peripartum and postpartum periods. Early TTS diagnosis in pregnancy is of great importance in improving both maternal and fetal mortality. Although TTS involves many pathogenetic pathways, the imbalance between declining estrogen and arising sympathetic nervous system tone plays an important role. This review aims to provide recent published evidence of TTS in pregnancy and delve into the epidemiology of TTS in pregnancy, the pathophysiological mechanisms involved, the prognosis of TTS for the mother and the fetus, and its therapeutic multi-disciplinary management.

## 1. Introduction

Takotsubo syndrome (TTS) was first reported in 2017 by Pelliccia et al., describing the appearance of the left ventricle (LV) in an echocardiogram at the end of the systolic phase that resembles the Japanese octopus trap (tako for octopus, tsubo for pot) [1]. Takotsubo syndrome, also known as stress-induced cardiomyopathy or “broken heart syndrome,” is characterized by the acute but temporary abnormality of the LV’s motility in the absence of acute coronary syndrome (ACS) [2,3,4]. Although the term “cardiomyopathy” has been vastly used to refer to this condition, currently, the term “syndrome” is mainly utilized, as there is a lack of evidence of intrinsic myocardial structural pathology along with a high variety of accompanying clinical signs and symptoms, concomitant diseases, risk factors and prognostic indices. It is important to add that TTS mainly presents after a significant emotional- or physical- stress trigger factor, leading to alternative designations such as “broken heart syndrome” or “stress cardiomyopathy” [5]. A strong association with “negative” emotions was vividly described in the registry by Itoh et al., as they discovered a heightened prevalence of TTS in the Japanese population after the Great East Japan Earthquake [6]. Acute physical stress and emotional stress seem to be important triggers of this medical condition, affecting mainly postmenopausal women between 60 and 70 years old [7,8]. Recently, it has been demonstrated that acute stress not only affects the ventricular myocardium and the vasculature tone has been associated with transient changes in the severity of mitral valve disease [9]. TTS during pregnancy or the immediate postpartum period is shown to be very rare but significant for maternal and fetal morbidity. Furthermore, the differential diagnosis of TTS includes peripartum cardiomyopathy, and the distinct recognition of both entities, though rare, is of great importance in both maternal and fetal outcomes [10]. Both entities represent a crucial diagnostic and therapeutical challenge, as they are potential causes of heart failure development in women [11]. In our review, we aim to outline the recently published evidence of TTS in pregnancy and delve into the epidemiology of TTS in pregnancy, the pathophysiological mechanisms involved, the prognosis of TTS for the mother and the fetus, and its therapeutic multi-disciplinary management.

## 2. Definition and Epidemiology

Several diagnostic criteria have been proposed for the diagnosis of TTS. According to the revised Mayo Clinic criteria, they include LV midsegment transient dyskinesia; regional wall motion abnormalities beyond a single epicardial vascular territory; absence of obstructive coronary artery disease/plaque rupture; new electrocardiographic abnormalities or modest troponin elevation; and absence of pheochromocytoma (Table 1) [12]. Moreover, the latest consensus from the European Society of Cardiology described the International Takotsubo Diagnostic Criteria (Table 2) [12,13,14,15].

Among the 5–6% of women presenting with suspected ST-elevation myocardial infarction (STEMI), only 1–3% of cases are considered to be due to TTS [16,17,18]. Many studies have shown that 90% of the patients with TTS are women between 67 and 70 years old, and 80% of them are over 50 years old [2,19]. Specifically, women over 55 years old seem to have a 10-fold higher risk for TTS than men [20].

It is known that TTS shares common prognostic characteristics with acute coronary syndromes (ACSs) [21]. Regarding traditional cardiovascular risk factors, hypertension is comparably represented in TTS and ACS patients (about 55%), while, on the other hand, diabetes mellitus and chronic kidney disease have lower prevalences (21% and 10%, respectively) [22]. Similarly, findings from the Tokyo Cardiovascular Care Unit network registry, published by Isogai et al., compared TTS and patients with ACS, reporting lower rates of diabetes (17% vs. 28%) and hypercholesterolemia (25% vs. 43%) [23]. Consistent results have been demonstrated in the InterTAK registry, the SWEDHEART registry, the GEIST registry, and the RETAKO registry [24,25,26,27]. In pregnancy, only 2% of women are shown to suffer from cardiovascular complications, and pregnancy-associated TTS and cardiomyopathies during this period of a woman’s life represent a rare diagnosis [3]. Due to the similarities between TTS and peripartum cardiomyopathy (PPCM), there is endogenous difficulty in precisely detecting the prevalence of TTS in pregnant women. Vazirani et al. evaluated the characteristics of patients with peripartum TTS from the RETAKO registry (n = 6), showing that the mean age of the population was 38.3 ± 2.3, one patient had hypertension (16.7%), one patient had dyslipidemia (16.7%), one patient had a history of smoking (16.7%), and there were no patients with diabetes or obesity [28]. Moreover, the researchers compared their results to peripartum TTS patients from the literature (n = 32); the small size of these populations cannot lead to reliable results or extrapolation regarding not only epidemiological but also clinical and echocardiographic characteristics, but it highlights the strong need for registries, even multi-national, that require co-operation from several different specialties for this special population.

## 3. Pathophysiology

The exact pathogenetic mechanisms leading to TTS are still unknown. Various hypotheses for its pathophysiology have been proposed, but none provide a complete explanation of the syndrome [29]. Numerous clinical studies have tried to explain the cardiovascular response in situations of extreme stress [30,31]. Two physiological aspects are important for understanding the syndrome. The first is the activation of the centers of the brain and hypothalamic-pituitary-adrenal axis in stress situations and the levels of released epinephrin and norepinephrine. The second is the response of the cardiovascular and autonomous neural system to the above-mentioned changes [26,32]. Wittstein et al. showed that the levels of catecholamines in TTS patients were higher than in the general population and patients with acute left anterior descending (LAD) artery occlusion and ST-segment elevation myocardial infarction (STEMI) [33]. Furthermore, pheochromocytoma, acute subarachnoid hemorrhage with sympathetic storm, or acute thyrotoxicosis, clinical conditions characterized by elevated catecholamine levels, can lead to TTS [34,35,36]. Abraham et al. demonstrated that TTS could be triggered after administration of adrenaline or dobutamine, supporting the causative role of catecholamines in TTS [37].

The different inotropic responses of the heart’s β-adrenergic receptors (βAR1 and 2) to catecholamines seem to play a significant role in the pathophysiology of TTS. There has been extensive research on the involvement of β-adrenergic receptors (βARs) towards unraveling the truth in the underlying pathophysiology of regional wall abnormalities [38]. It is of great importance to understand that βARs are not equally distributed in the myocardium, with the highest concentration being in the apex [39,40]. Although adrenaline and noradrenaline cause a positive inotropic reaction via the β1AR-Gs pathway, they influence the β2AR differently. Furthermore, the apical myocardium is exposed to circulating adrenaline stimuli, while basal parts with greater density of sympathetic nerve terminals are stimulated by noradrenaline [39,40]. All doses of noradrenaline and low/medium doses of adrenaline lead to a positive inotropic reaction via the β2AR-Gs pathway, while high doses of adrenaline activate the negative inotropic β2AR-Gi pathway [41]. The hypercontractility of the apical myocardium, followed by hypokinesia during acute stress situations, can be explained by this physiological mechanism [42]. Although apical ballooning is the most common echocardiographic presentation of TTS in pregnant women, there are reports in the literature of reverse TTS during pregnancy, characterized by basal ballooning and hypercontractility of the apex [43].

Multivessel vasospasm and its role in TTS have also been investigated by many experts. Although in the majority of cases no vasospasm is observed, there are some cases, especially during the acute phase of the syndrome, in which this phenomenon is reported [44,45]. Regarding acute atherosclerotic plaque rupture, there is no significant evidence showing a strong relationship with TTS [30,46]. Furthermore, estrogens seem to play a causative role in the pathophysiology of TTS. Specifically, they regulate the sympathetic tone via βAR expression in the cardiovascular system, which leads to a tonic-relative suppression of the receptors in women during their fertile years. However, this effect diminishes in menopause, resulting in elevated myocardial and vascular responses to βAR agonists, which explains the high prevalence of the syndrome in postmenopausal women [47]. TTS in obstetric patients can be a critical medical condition for both the mother and the fetus. Temporary coronary vasospasm is thought to also play a causative role [13,26]. This endothelial impairment is followed by reduced blood flow through the uteroplacental unit, resulting in pregnancy-related complications such as preeclampsia [48]. In addition to induced alterations, the acute decrease in maternal cardiac output leads to an increase in the systematic vascular resistance, causing reduction in both uterine and fetal blood supply, directly influencing the fetal outcomes of those pregnancies [49].

Massive psychological and physical stress during pregnancy and delivery (vaginal delivery—VD—or cesarean section—CS) with acute release of catecholamines and hemodynamic changes seems to be of great importance in the pathophysiology of TTS [50]. Specifically, pregnancy is considered to be a “9-month-long cardiovascular stress test” due to the normal hemodynamic changes that occur during it [51]. These cardiovascular alterations begin in the first weeks of the pregnancy and especially at 5 weeks of gestational age [52]. In particular, hormonal changes in the first trimester, such as high levels of estrogen, progesterone, and relaxin, seem to lead to systemic vasodilation and decrease in peripheral vascular resistance up to 25–30% [53]. This decrease reaches its lowest point during the second trimester, then stabilizes for the rest of the pregnancy and returns to the starting situation postpartum [53,54,55]. As a compensatory mechanism of these physiological modifications, cardiac output increases sharply (40%) during the first trimester and also continues to rise during the second one [56,57]. Cardiac output in twin pregnancies seems to rise 15% more than in a singleton pregnancy [58]. Blood pressure is also influenced during this period, with a decrease in systolic blood pressure (SBP), diastolic blood pressure (DBP), mean arterial pressure, and central SBP being detected, especially in the first trimester. This is also the reason why cardiovascular changes in pregnant women should always be compared with periconceptional measurements [53]. Heart rate, on the other hand, seems to increase during the whole pregnancy [53,58]. During labor, cardiac output increases up to 80%, and all these hemodynamic changes reset completely 6 months after delivery (Figure 1). Nevertheless, medication applied pre- or intra-partum is shown to be associated with the onset of TTS, especially in patients undergoing CS. The most frequently used medications proven to induce cardiovascular changes leading to TTS are oxytocin, ritodrine, isoxsuprine, ergovine, methylergometrine, sulprostone, and ephedrine [4,59,60,61,62,63]. In summary, the exact pathophysiological mechanisms elucidating the clinical characteristics of TTS are still unknown and need more investigation to be thoroughly clarified.

## 4. Clinical Presentation and Differential Diagnosis

### 4.1. Clinical Presentation

The most common symptoms of TTS in the general population are acute chest pain, dyspnea, and syncope. However, patients can be asymptomatic, and TTS can be diagnosed by new ECG changes or even abrupt elevation of cardiac markers [2]. TTS in obstetric women is a rare but life-threatening clinical condition that can emerge during various stages of pregnancy and the peri- and postpartum periods and appear with various symptoms. In the largest published cohort with TTS in peripartum, Vazirani et al. showed that patients with peripartum TTS show a higher incidence of dyspnea with atypical symptoms and echocardiographic patterns and less frequent ST-segment elevation findings. Nevertheless, peripartum TTS patients seemed to have a higher Killip status and a lower LVEF on admission but experience excellent mid- and long-term prognosis following the acute phase [28]. Udemgba et al. demonstrated that women undergoing CS and having a greater proportion of anesthesia may develop a postpartum TTS with an earlier onset compared with women who deliver vaginally [64]. However, no significant difference was mentioned in the degree of systolic dysfunction between the two groups [64].

Mogos et al. investigated in a retrospective cross-sectional analysis the prevalence of pregnancy-associated TTS hospitalizations in the United States and showed that pregnant women with advanced maternal age, history of tobacco and opioid use, and cardiovascular/cerebrovascular disease comorbidities, as well as pregnancy-related morbidities, were more likely to develop TTS. Furthermore, TTS seems to arise most often during postpartum hospitalization compared to antepartum and delivery hospitalizations. It was also demonstrated that hospitalizations complicated by TTS were strongly related to a higher risk of in-hospital mortality and extended hospital stays [65]. Therefore, it is highlighted in the literature that TTS can occur not only antepartum, but also peripartum, and there should be a high alert for up to 12 months after pregnancy [66].

### 4.2. Differential Diagnosis

Beyond clinical presentation, ECG changes, cardiac biomarkers, and transthoracic echocardiogram (TTE) contribute significantly in the diagnostic algorithm of TTS. Regarding ECG changes at admission, TTS often mimics acute coronary syndromes, more often presenting with ST-depression/elevation (ST changes rarely involve V1 lead), QTc prolongation, and T-wave inversion and can resemble left anterior descending artery myocardial infarction (Figure 2) [67]. It is important to note that ST-segment depression is more frequent in inferior leads, and subsequent normalization of the ST segment associated with a ubiquitous T-wave inversion pattern is regularly observed [68]. Another apparent characteristic of ECG in TTS patients is that QT interval prolongation is presented without pathological Q waves [68,69]. Contrary to typical ACS, ST-segment depression or more profound elevation is not associated with the prediction of ventricular arrhythmias in TTS [70]. Multimodality imaging is of great importance, even in pregnancy and postpartum periods, for the diagnosis of TTS—in Figure 3, typical imaging characteristics of TTS in CMR are presented.

Diagnosis of TTS can be tricky due to its common clinical features with peripartum cardiomyopathy [71]. Peripartum cardiomyopathy was first described by Demakis et al. in 1971 [72]. Both clinical conditions are related with pregnancy and lead to left ventricular dysfunction with acute heart failure symptoms, and left ventricular contractility usually recovers at follow-up examination [73,74]. However, it is of great importance to differentiate between these two medical conditions (Table 3), as TTS seems to have a better prognosis and entirely different management compared to peripartum cardiomyopathy. Regarding clinical characteristics, TTS seems to be associated with physical or sentimental trigger factors and characterized by acute and sudden onset, while on the other hand, peripartum cardiomyopathy is characterized by gradual presentation, usually with complete absence of triggering factors [71,74].

Segmentary involvement with well-described patterns characterizes peripartum TTS, while PPCM patients typically appear to have LV dilation with global dysfunction [74]. Furthermore, peripartum TTS arises more often a few days ante- or postpartum and is correlated with a higher elevation of troponin. On the other hand, peripartum cardiomyopathy occurs most likely at the end of the pregnancy or in the first months postpartum [73,74,75]. These two entities can be distinguished during follow-up. Most patients with peripartum TTS recover after one month, while 86.9% of patients with PPCM continue to experience persistent heart failure at one month, with only 46% recovering left ventricular function by six months [76,77,78].

### 4.3. Fetal Outcomes

Despite the great impact of TTS on a woman’s life, it is shown that this clinical condition could also affect the fetus. In particular, Udemgba et al. showed that worse fetal outcomes were detected when the onset of TTS was during the pregnancy rather than postpartum [64]. Indeed, healthy baby deliveries were more frequent in postpartum TTS (45.7%) compared to pregnancy TTS (7.4%). This is supported by the acute nature of the disease, implying that there were fewer or no abnormal cardiovascular alterations earlier in the pregnancy for women with TTS in the postpartum period. The correlation between TTS and the fetus can be explained through endothelial dysfunction and vasospasm, causing reduced blood flow through the uteroplacental unit, which can lead to vascular complications during pregnancy such as preeclampsia [48]. Furthermore, in pregnant women with TTS, cardiac output decreases abruptly and systematic vascular resistance increases, resulting in impaired blood supply to the uterus and the fetus [49].

### 4.4. Complications and Recurrence Rate

Patients with pregnancy-associated TTS have been shown to fully recover after a few months. However, various complications of TTS during pregnancy, including ventricular tachycardia, ventricular fibrillation (VF), AV block, and long QTc have been reported. Lee et al. have presented a case of a woman during the postpartum period with persistent conduction blocks, by whom a pacemaker insertion was obligatory [79]. El-Battrawy et al. investigated in a multicenter international registry trial the relapse rate of pregnancy-associated TTS and reported a 1–11% recurrence rate. It was also demonstrated that the younger population has an elevated risk for relapse [80]. Although the recurrence rate of TTS occurring during pregnancy is rather rare, it is reasonable that patients require close follow-up with short-term visits after discharge, as a small percentage of women may deteriorate, either postpartum or later in the first year, presenting with various cerebrovascular or thromboembolic events [2,66,81].

## 5. Management of TTS

The therapeutical strategy for TTS, outside pregnancy or the postpartum setting, is mainly based on retrospective or observational data, as there is a scarcity of randomized evidence, highlighting the need for the primum non nocere (“first, do no harm”) approach. This is more profound in pregnancy-associated TTS, where there is a need to classify treatment options that are not only safe and efficient for the woman but also safe for the fetus. This is presented in Figure 4 and described in the forthcoming section.

The primary management of pregnancy-associated TTS is mainly based on the diminishment of the emotional or physical stress that triggered this clinical entity. Women presenting with symptoms of acute myocardial infarction should receive morphine, oxygen, and aspirin [82]. However, morphine can be threatening for the newborn due to causing respiratory depression when it is applied close to the delivery. This is a very important point that should be considered while managing peripartum TTS [83]. Regarding aspirin and all other antiplatelets, more multicenter clinical trials are needed to investigate the role of their short- and long-term usage in patients without coronary artery disease [84].

In our therapeutic arsenal for managing acute heart failure caused by TTS, we have angiotensin enzyme inhibitor (ACEI), angiotensin II receptor blocker (ARB), beta-blockers, diuretics, and nitroglycerin, although the first two are contraindicated during pregnancy due to their correlation with fetal anomalies [85]. However, ACEIs are reported to have been used in numerous cases of TTS postpartum, especially when left ventricular function deteriorates. Beta blockers, on the other hand, can be used with precaution in cases of bradycardia or prolonged QT interval [86,87,88,89].

Despite acute heart failure, women with pregnancy-associated TTS can develop acute cardiac collapse, for which the therapeutic approach depends on the coexistence of left ventricular outflow tract obstruction (LVOTO). Fluid replacement in combination with beta blockers is of great importance for the management of this clinical condition, after ruling out the presence of pulmonary edema. On the other hand, inotropes such as adrenaline and noradrenaline should be restricted only in situations involving cardiac arrest or shock, as they can trigger TTS [40,63]. In cases with LVOTO, oxygen supplementation, diuretics, and carperitide should be used for controlling the pulmonary edema [90,91]. Furthermore, levosimendan is a calcium sensitizer that can be used as the initial step in transitioning away from inotropes. It has been successfully used in cases of TTS, especially peripartum [92,93]. On the other hand, bromocriptine, which belongs to the therapeutic arsenal of PPTM, seems to have a vasoconstrictive role, which can eventually lead to TTS, highlighting its contraindication in such a setting [94,95]. Udemgba et al. demonstrated in a scoping literature review that short-term mechanical support has been used in 11 patients (utilization of IABP in 6 patients in postpartum cardiomyopathy and ECMO in 3 patients during pregnancy and 2 patients in the postpartum period). Larger registries that include this special population are more than needed for better understanding the extrapolative nature of treatment performed [64].

During the acute phase of pregnancy-associated TTS, it is more likely for patients to develop thrombotic or thromboembolic outcomes. In particular, women with LV thrombus can be treated with oral anticoagulants for three months and typically show a full recovery after this period of time [96,97]. Pregnancy and postpartum periods are characterized by hypercoagulability and a high risk of thromboembolism, especially in adipose women, women older than 35 years old, or those with hospitalization longer than three days. Thus, it seems vital for pregnant women suffering from TTS to be anticoagulated. Unfractionated and low-molecular-weight heparin are considered safe during pregnancy and can be used in those cases, with a simultaneous evaluation of bleeding risk and need for epidural anesthesia or urgent operations [98].

## 6. New Pregnancy After TTS—How to Manage It?

In case of a subsequent pregnancy after TTS, it is of great importance to prevent the recurrence of this life-threatening condition with a multidisciplinary team from the beginning of the pregnancy. The idea of a multidisciplinary team, the pregnancy heart team, was introduced by the 2018 ESC Guidelines for the management of cardiovascular diseases during pregnancy for those at moderate or high risk of complications during pregnancy (mWHO II–III, III, and IV) [99]. Furthermore, the team should at least consist of a cardiologist, an obstetrician, and an anesthetist with high expertise and experience with the treatment of high-risk pregnant women. Fully detailed informed counseling of the risks related to the pregnancy, as well as clinical and echocardiographic follow-up visits that include estimation of the LV systolic function and hemodynamics’ monitoring, are of great importance. Moreover, the preservation of the mental well-being of the women is vital during the whole pregnancy [80].

To date, there is no consensus regarding the time and the mode of delivery for these patients. Our primary aim is the maintenance of low catecholamines levels, with pregnancy-allowed b-blockers, as well as the avoidance of both regional and systematic sympathomimetic agents. Regarding anesthesia, an epidural should be favored over a subarachnoid block to avoid rapid sympathectomy and subsequent hypotension [80]. During the long-term treatment of anticoagulation treatment in patients, with slow recovery of the left ventricle and concomitant anticoagulation treatment, there should be increased caution with a minimum amount of time between the last dose of antithrombotic agents and the epidural anesthesia [100,101].

## 7. Conclusions

Pregnancy-associated TTS is a not-so-rare, life-threatening medical condition that remains under-recognized. Strong suspicion and understanding of the disease’s diagnosis, progression, and management seem to be crucial in reducing maternal morbidity and mortality. Differentiating TTS from other cardiomyopathies such as PPTM is vital. However, more multicenter registries are needed to understand and investigate the pathophysiology of the disease and to establish consensus documents and recommendations for the diagnosis and the management of pregnancy-associated TTS.

## Figures and Tables

**Figure 1 jcm-14-02356-f001:**
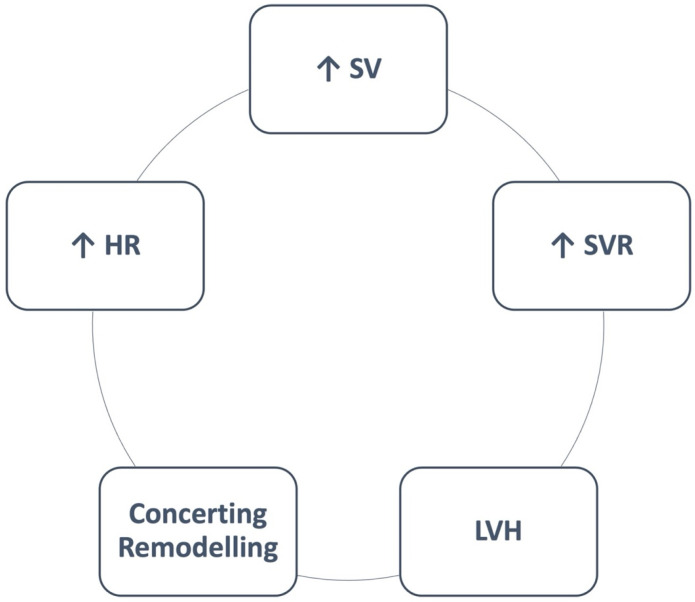
**Cardiovascular changes during pregnancy.** HR: heart rate, LVH: left ventricular hypertrophy, SVR: systematic vascular resistance, SV: stroke volume.

**Figure 2 jcm-14-02356-f002:**
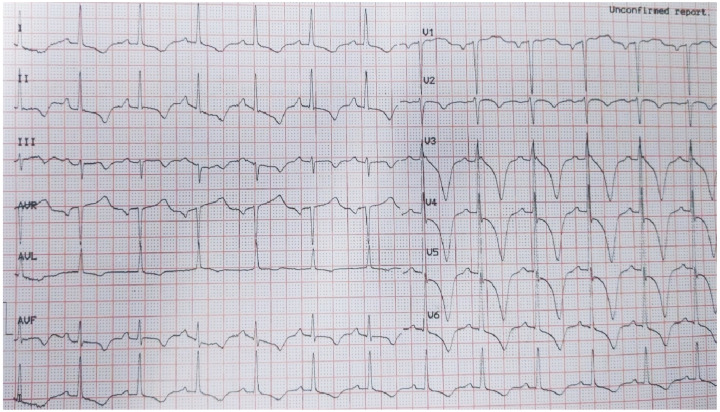
**ECG of a female patient with TTS during postpartum.** Widespread T-wave inversion together with ST depression in lateral precordial leads alongside marked QT prolongation.

**Figure 3 jcm-14-02356-f003:**
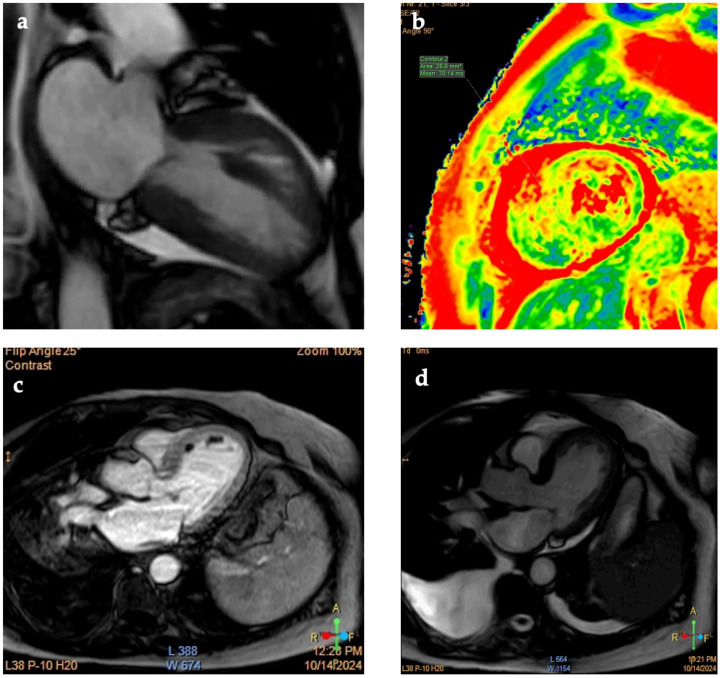
**CMR findings in TTS.** (**a**) CINE 2-chamber view with typical TTS—apical ballooning; (**b**) T2 mapping edema accompanies TTS in acute-subacute phase (increased time in T2 mapping); (**c**) CMR can detect TTS complications such as LV thrombi or pericardial/pleural effusion seen in next picture; (**d**) LGE absence is a common finding in TTS and, if present, is commonly attributed to generalized edema, which dissolves quickly.

**Figure 4 jcm-14-02356-f004:**
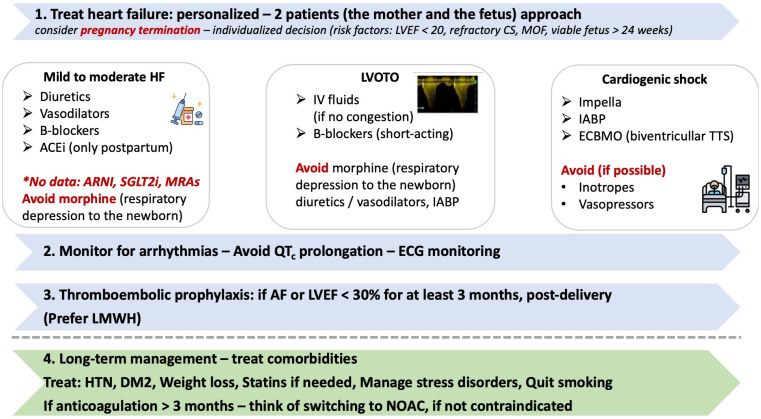
**Therapeutical strategies in TTS during pregnancy or postpartum period.** ACEi: angiotensin-converting enzyme inhibitors, AF: atrial fibrillation, ARNI: angiotensin receptor/neprilysin inhibitor, CS: cardiogenic shock, DM2: diabetes mellitus type 2, ECG: electrocardiogram, ECMO: extracorporeal membrane oxygenation, HF: heart failure, HTN: hypertension, IABP: intra-aortic balloon pump, LMWH: low molecular weight heparin, LVEF: left ventricular ejection fraction, LVOTO: left ventricular outflow tract obstruction, MOF: multiple organ failure, MRA: mineralocorticoid receptor antagonist, NOAC: non-vitamin K oral anticoagulants, SGLT2i: sodium-glucose cotransporter-2 inhibitor.

**Table 1 jcm-14-02356-t001:** Mayo Clinic criteria for the diagnosis of takotsubo cardiomyopathy [12].

Transient hypokinesis, akinesis, or dyskinesis of the left ventricular mid-segments with or without apical involvement; the regional wall motion abnormalities extend beyond a single epicardial coronary distribution; a stressful trigger is often but not always present.
2. Absence of obstructive coronary disease or angiographic evidence of acute plaque rupture.
3. New electrocardiographic abnormalities (either ST-segment elevation and/or T-wave inversion) or modest elevation in cardiac troponin.
4. Absence of phaeochromocytoma, myocarditis.

**Table 2 jcm-14-02356-t002:** InterTAK criteria for the diagnosis of takotsubo cardiomyopathy [13].

Patients show transient left ventricular dysfunction (hypokinesia, akinesia, or dyskinesia) presenting as apical ballooning or midventricular, basal, or focal wall motion abnormalities. Right ventricular involvement can be present. Besides these regional wall motion patterns, transitions between all types can exist. The regional wall motion abnormality usually extends beyond a single epicardial vascular distribution; however, rare cases can exist where the regional wall motion abnormality is present in the subtended myocardial territory of a single coronary artery (focal TTS).
2. An emotional, physical, or combined trigger can precede the takotsubo syndrome event, but this is not obligatory.
3. Neurologic disorders (e.g., subarachnoid haemorrhage, stroke/transient ischaemic attack, or seizures) as well as pheochromocytoma may serve as triggers for takotsubo syndrome.
4. New ECG abnormalities are present (ST-segment elevation, ST-segment depression, T-wave inversion, and QTc prolongation); however, rare cases exist without any ECG changes.
5. Levels of cardiac biomarkers (troponin and creatine kinase) are moderately elevated in most cases; significant elevation of brain natriuretic peptide is common.
6. Significant coronary artery disease is not a contradiction in takotsubo syndrome.
7. Patients have no evidence of infectious myocarditis.
8. Postmenopausal women are predominantly affected.

**Table 3 jcm-14-02356-t003:** Differential diagnosis between peripartum cardiomyopathy and takotsubo syndrome.

	Peripartum Cardiomyopathy	Takotsubo Syndrome
**Onset**	Weeks around delivery	Hours to days around delivery
**Biomarkers**	Mild elevation of serum troponin Other: BNP, CRP, microRNA-146a, cathepsin D	Higher elevation of serum troponin Other: BNP, CRP
**ECG**	Normal, ST depression, T-wave abnormalities, 2nd- or 3rd-degree atrioventricular block, RBBB	ST depression/elevation (ST changes rarely involve V1 lead) QTc prolongation, T-wave inversion
**TTE**	Left ventricle dilatation with global hypokinesia Left (or biatrial) enlargement	Regional left ventricular dysfunction with apical hypo- or akinesia
**CMR**	High-signal T2 (edema) Global/regional wall motion abnormalities Non-specific distribution of LGE	High-signal T2 (edema) Regional wall motion abnormalities Absence (usually) of LGE in the acute stage
**Recovery time**	34% LVEF recovery at 6 months 47% LVEF recovery at 12 months Mortality rate: 1.6–27.6%	Almost full LVEF recovery in 60 days Persistent LVEF reduction (comorbidities) Mortality rate: 4.5–5.6%

AF: atrial fibrillation, BNP: B-type natriuretic peptide, CMR: cardiac magnetic resonance imaging, CRP: C-reactive protein, LGE: late-gadolinium enhancement, LVEF: left ventricular ejection fraction, TTE: transthoracic echocardiogram.

## Data Availability

As this is a review article, no data were created.

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
