# Peer review of "Pregnancy-Associated Takotsubo Syndrome: A Narrative Review of the Literature"

_jcm, 2025, doi:10.3390/jcm14072356_

Round 1
Reviewer 1 Report
Comments and Suggestions for Authors
The authors, in their paper entitled "Pregnancy associated Takotsubo Cardiomyopathy: a narrative review of the literature" provided an interesting overview regarding the complex clinical scenario of takotsubo syndrome during pregnancy.
The review is interesting and well structured; however, I have few observations to improve the overall quality of the paper
1) I wonder why the authors chose to use the term "Takotsubo cardiomyopathy" in the title when they clearly state that it should be mentioned as syndrome
2) Please provide some ECG, echo and CMR examples of TTS in pregnancy to implement further the iconography of the paper
3) The differential diagnosis section is poor: I suggest to implement a more detailed description of the main findings at cardiac ultrasound and CMR of TTS and peri-partum cardiomyopathy
4) A table regarding the multifaceted therapeutical strategies to TTS would be of additional value
5) In the introduction, the authors should mention that TTS during pregnancy and peripartum cardiomyopathy represent a diagnostic challange as they are potential causes of heart failure in women (please refer to this interesting review: PMID: 37504533)
Comments on the Quality of English LanguageEnglish is fine
Author Response
Reviewer 1
The authors, in their paper entitled "Pregnancy associated Takotsubo Cardiomyopathy: a narrative review of the literature" provided an interesting overview regarding the complex clinical scenario of takotsubo syndrome during pregnancy.
The review is interesting and well structured; however, I have few observations to improve the overall quality of the paper
1) I wonder why the authors chose to use the term "Takotsubo cardiomyopathy" in the title when they clearly state that it should be mentioned as syndrome
2) Please provide some ECG, echo and CMR examples of TTS in pregnancy to implement further the iconography of the paper
3) The differential diagnosis section is poor: I suggest to implement a more detailed description of the main findings at cardiac ultrasound and CMR of TTS and peri-partum cardiomyopathy
4) A table regarding the multifaceted therapeutical strategies to TTS would be of additional value
5) In the introduction, the authors should mention that TTS during pregnancy and peripartum cardiomyopathy represent a diagnostic challange as they are potential causes of heart failure in women (please refer to this interesting review: PMID: 37504533)
Reply to Reviewer 1
We would like to thank the Reviewer for the kind comments and suggestions and we have implemented them in our manuscript. We reply point-by-point to each comment.
- It is of great importance that you have noted this discordance and we have changed the title by using the term Takotsubo syndrome (TTS) in the title, instead of Takotsubo cardiomyopathy.
2) We have added:
- a figure (Figure 2) with an ECG of a female patient with TTS during postpartum
- a figure (Figure 3) with typical findings of TTS syndrome in CMR
3) Thank you for your kind comment – we have described the differential diagnosis of the main findings (TTE, CMR), as well as clinical characteristics and biomarkers rise in Table 2, and we did not want to repeat them in the manuscript. Due to the addition of more text, figures and the therapeutic algorithm, we focus on the differential diagnosis on the Table. However, we have enriched the text, referring to the Table.
4) We have added an introductory paragraph (highlighted) in the “Management section”, as well as figure (Figure 4), regarding therapeutical strategies in pregnancy-associated TTS.
5) We have implemented your comment in the introduction, as well as adding the suggested reference (highlighted text in Introduction section).
Reviewer 2 Report
Comments and Suggestions for Authors
In this interesting paper, the authors described the epidemiology, pathophysiology, clinical presentation and management of TTS in pregnancy.
Massive psychological and physical stress during pregnancy and delivery may cause acute release of catecholamines, thus contributing to TTS occurrence.
Worse fetal outcomes occur when TTS onset is during pregnancy, rather than postpartum.
TTS postpartum has been treated with ACEi and/or Beta blockers, generally contraindicated during pregnancy.
The manuscript is well written, the tables are clear, each section is exhaustively presented, the references are appropriate and the conclusions correctly summarize the main findings of the review.
I have only one suggestion for the authors.
It is important the differential diagnosis between TTS and peripartum cardiomyopathy. This topic is important for clinical cardiologists.
In the Paragraph 4.2. "Differential diagnosis", the authors could mention more ECG criteria compatible with TTS, particularly ST elevation in both inferior and anterior leads except in
V1, absence of reciprocal ST-segment depression in inferior leads, subsequent normalization of ST segment associated with ubiquitous T-wave inversion pattern, and corrected QT interval prolongation without pathological Q waves, all typical ECG changes observed in TTS (PMID: 33482618 and PMID: 37581859).
In the Table 2, the authors could specify that ST abnormalities do not involve V1 lead.
Author Response
Reviewer 2
In this interesting paper, the authors described the epidemiology, pathophysiology, clinical presentation and management of TTS in pregnancy.
Massive psychological and physical stress during pregnancy and delivery may cause acute release of catecholamines, thus contributing to TTS occurrence.
Worse fetal outcomes occur when TTS onset is during pregnancy, rather than postpartum.
TTS postpartum has been treated with ACEi and/or Beta blockers, generally contraindicated during pregnancy.
The manuscript is well written, the tables are clear, each section is exhaustively presented, the references are appropriate and the conclusions correctly summarize the main findings of the review.
I have only one suggestion for the authors.
It is important the differential diagnosis between TTS and peripartum cardiomyopathy. This topic is important for clinical cardiologists.
In the Paragraph 4.2. "Differential diagnosis", the authors could mention more ECG criteria compatible with TTS, particularly ST elevation in both inferior and anterior leads except in
V1, absence of reciprocal ST-segment depression in inferior leads, subsequent normalization of ST segment associated with ubiquitous T-wave inversion pattern, and corrected QT interval prolongation without pathological Q waves, all typical ECG changes observed in TTS (PMID: 33482618 and PMID: 37581859).
In the Table 2, the authors could specify that ST abnormalities do not involve V1 lead.
Reply to Reviewer 2
We would like to thank the Reviewer for the kind comments and recommendations.
- We have added a section (highlighted changes in paragraph 4.2 with ECG criteria in TTS), as well as adding the suggested references (PMID: 33482618 and PMID: 37581859).
- It is of great importance mentioning the lack of ST abnormalities in V1 lead. We added in Table 2 that “ST abnormalities do not involve V1 lead”, as well as in the text, adding these references:
- https://pmc.ncbi.nlm.nih.gov/articles/PMC9096129/
- https://pubmed.ncbi.nlm.nih.gov/36074030/
Round 2
Reviewer 1 Report
Comments and Suggestions for Authors
I kindly thank the authors for the effort to improve their paper. No additional observations.